# AMSCN: A Novel Dual-Task Model for Automatic Modulation Classification and Specific Emitter Identification

**DOI:** 10.3390/s23052476

**Published:** 2023-02-23

**Authors:** Shanchuan Ying, Sai Huang, Shuo Chang, Jiashuo He, Zhiyong Feng

**Affiliations:** Key Laboratory of Universal Wireless Communications, Ministry of Education, Beijing University of Posts and Telecommunications, Beijing 100876, China

**Keywords:** specific emitter identification (SEI), automatic modulation classification (AMC), deep learning, multitask learning

## Abstract

Specific emitter identification (SEI) and automatic modulation classification (AMC) are
generally two separate tasks in the field of radio monitoring. Both tasks have similarities in terms of
their application scenarios, signal modeling, feature engineering, and classifier design. It is feasible
and promising to integrate these two tasks, with the benefit of reducing the overall computational
complexity and improving the classification accuracy of each task. In this paper, we propose a
dual-task neural network named AMSCN that simultaneously classifies the modulation and the
transmitter of the received signal. In the AMSCN, we first use a combination of DenseNet and
Transformer as the backbone network to extract the distinguishable features; then, we design a
mask-based dual-head classifier (MDHC) to reinforce the joint learning of the two tasks. To train
the AMSCN, a multitask cross-entropy loss is proposed, which is the sum of the cross-entropy
loss of the AMC and the cross-entropy loss of the SEI. Experimental results show that our method
achieves performance gains for the SEI task with the aid of additional information from the AMC
task. Compared with the traditional single-task model, our classification accuracy of the AMC is
generally consistent with the state-of-the-art performance, while the classification accuracy of the SEI
is improved from 52.2% to 54.7%, which demonstrates the effectiveness of the AMSCN.

## 1. Introduction

Radio monitoring helps spectrum management agencies to plan and use frequencies [1], avoid incompatible uses [2], and identify sources of harmful interference [3]. With the development of wireless communication, ensuring the security of the communication process has become a topic of great concern [4,5]. A typical scenario is that in satellite communications, the ground station needs to receive signals relayed by satellites or other ground stations. In some civil–military satellite systems, both military satellites and civilian relay satellites may access the core network through the ground station, so the ground station needs to have the ability to discriminate radio signals to prevent interfering satellites or hostile satellites from accessing the network. If the authorization is based on the secret key, the ground station will face huge pressure when signal resolving because of the large number of signals attempting to gain access, and in many cases, the ground station is only responsible for relaying, and does not have the ability to resolve the signal content. Therefore a physical-layer-security-based authentication scheme is feasible. In addition, as satellite Internet technology is put into practice, more and more countries need to regulate the satellite networks of other countries, including service hours, service bands, radio signal range, and communication traffic. Specific emitter identification (SEI) and automatic modulation classification (AMC) are two common identification tasks for signal characteristics. SEI is a process of extracting individual characteristics from the signals and identifying the communication transmitters. These individual characteristics of a transmitter are often referred to as radio frequency (RF) fingerprints [6,7,8,9]. AMC is a process of blindly identifying the modulation format of an unknown received signal [10,11,12]. In the aforementioned communication scenario, the transmitter may change the current modulation at any time, and the receiver will be disturbed by the change in modulation when performing RF fingerprint recognition on the current signal. How to improve the efficiency of RF fingerprint recognition in the case of variable modulation becomes particularly important.

Generally speaking, the generalized modulation of signals is a process that makes certain characteristics of one waveform change according to another waveform or signal. The instantaneous amplitude, instantaneous phase, and other information of the baseband signal are not only artificially changed by the message signal, but they are also unintentionally modulated by the defects of the RF hardware. Therefore, both the AMC task and the SEI task can be viewed as the process of identifying the generalized modulation information from the received signal. However, in the existing research, these two tasks have been conducted separately and have not been studied together. In some specific monitoring scenarios, the need to identify both the modulation format and the RF fingerprints of the signal at the receiver side may exist simultaneously, and if we can use one model for both tasks, then the classification efficiency of the system can be improved.

In our work, we believe that it is feasible to implement both the AMC task and the SEI task in a single model for two reasons. Firstly, both are classification tasks, and they both extract the distinguishable features from the received signal. Secondly, in terms of the model design, the network structure of the two tasks is quite similar, which indicates that we can use the same network to perform both tasks. Moreover, the modulation and transmitter characteristics are contained within the same segment of the signal, and they both have similar significant impacts on the waveform of the signal. In traditional single-task detection, if a model is only used to identify modulation, then variations in the transmitter characteristics in the signal are viewed as interference to the AMC, and vice versa. If we learn these two characteristics simultaneously in one model, i.e., using two labels to guide the learning process of the network, then both kinds of information are valid for the model, which can facilitate the model to better distinguish between the two tasks.

In this paper, we design a framework for AMC and SEI signal characteristics classification using a multitask learning approach to mine the correlations between them and improve the recognition efficiency of both tasks. The contributions of the paper are as follows:We propose an AMC-mask-based SEI Classification Network (AMSCN) for the AMC and SEI. To our knowledge, this is the first approach to consider these two classification tasks together;In the AMSCN, we design a multitask classification model based on deep learning, which consists of a backbone network and a mask-based dual-head classifier (MDHC). The backbone network has a DenseNet–Transformer structure, which is responsible for extracting discriminative features that can be adapted to different signal feature scales in both tasks;The MDHC consists of an AMC head and an SEI head. It can enhance the correlation between the two tasks through a mask mechanism and finally output the classification results of the two tasks. With the help of the MDHC, we are able to balance the learning process using only the sum of the cross-entropy losses of the two tasks;We generate a simulated dataset for the AMC and SEI tasks. Extensive experiments are carried out on this simulated dataset to demonstrate that the fusion of AMC and SEI can achieve better predictions than single-task learning. Furthermore, some contrast experiments have also been conducted to verify the effectiveness of each module in the AMSCN.

This paper is organized as follows. In Section 2, we discuss the common feature design and model design approaches in the AMC and SEI domains, as well as the design of multitask models in deep learning. In Section 3, we introduce the entire classification framework, including the system model and the signal model. Section 4 details the design of the core module and the training method of the AMSCN. The results in terms of the accuracy and the additional ablation experiments are shown in Section 5. Finally, the conclusion and the prospective research activities are presented in Section 6.

## 2. Related Work

Through the development of artificial intelligence (AI), AI-based applications have entered every aspect of human society, including industry [13], agriculture [14], healthcare [15], and education [16]. In wireless communications, deep-learning-based methods also introduced a new way to consider the problem of signal characteristics recognition [17,18,19,20,21,22,23,24,25]. For the AMC and SEI tasks, the deep-learning-based approach is essentially a statistical pattern recognition problem, which can be divided into two steps: feature extraction and pattern recognition [26]. That is, first, we extract the reference features from the received signal; then, we judge the modulation type or RF fingerprints based on these features. In addition to extracting the features based on time–domain waveforms [8,27,28,29], there are also preprocessing methods that convert waveforms into explicit features for deeper feature extraction, such as time–frequency diagrams [30], spectrograms [31], higher-order cumulants [32], wavelet transform features [33], cyclostationary features [34], constellation diagrams [23,35], etc [36,37]. These preprocessing methods are applicable to both AMC and SEI tasks almost simultaneously, indicating the similarity of the essential characteristics of the AMC and SEI tasks. In terms of the model design, the two tasks also share similarities; for example, structures, such as regular convolution neural networks (CNNs) [22,38], ResNet [37,39], Inception [12], DenseNet [40,41], long short-term memory (LSTM) [42], and Transformer [7,43], are present in both tasks. The model of one task can achieve good results on the other task after some simple modifications.

In deep learning, multitask learning has been widely researched. Multitask learning is a method that enables a model to have better generalization performance on the original task by sharing feature representations between related tasks [44]. The benefits of multitask learning are manifold; it not only saves network parameters through hard parameter sharing [45,46] but also reduces the overfitting of the model on a single task [47]. In the field of radio signal classification, there is no multitask learning model combining AMC and SEI, but some multitask learning applications exist in other scenarios. In [48], an algorithm to simultaneously learn the modulation method and signal-to-noise ratio was proposed, and the results showed that the addition of a new task improved the classification efficiency of the AMC. In [49], a multitask deep convolutional neural network was proposed to perform a modulation classification and direction-of-arrival (DOA) estimation simultaneously.

Apart from multitask learning, attention mechanisms are increasingly applied to signal detection. Ref. [43] used an R-transformer-based model to achieve state-of-the-art performance on the AMC task. Ref. [50] was inspired by the high-quality representation capability of Transformer and proposed a novel openset SEI algorithm to find accurate and stable boundary samples with more robust representations. In our practice, we found that the self-attention mechanism or Transformer structure could effectively capture the changes in global information in the case of longer sequences, and thus, it is more suitable than LSTM for the tasks of AMC and SEI, which have large differences in feature scales.

## 3. System Model and Problem Statement

In this section, we provide an assumption of the application scenario, the workflow of the whole framework, and the signal model of the simulated dataset.

### 3.1. System Model

We assumed that there was an application scenario where the transmitter had multiple similar devices that could establish communication with the receiver, and that these devices constantly changed the modulation when sending signals; meanwhile, the receiver only received signals from one device at the same time. In order to achieve the effect of safe signal accessing, the receiver needed to determine the modulation of the current signal for subsequent algorithm demodulation, as well as the individual transmitter to which the current signal belongs. Figure 1 shows the framework of our proposed multitask classification method. The receiver acquires the signal and transforms it into a complex baseband signal sequence x(n) with a length of *N*. The real part of each complex sampling point is called the in-phase (I) component, and its imaginary part is called the orthogonal (Q) component. Then, the proposed AMSCN is applied to predict both the modulation and the transmitter of the received signal, where $H_{i,j}$ indicates that the current signal uses the *i*th modulation format in the modulation set, together with the *j*th device in the transmitter set. Finally, these predictions can be used to support subsequent communication applications, such as demodulation and access authorization.

### 3.2. Signal Model

In a communication system, the transmitter sends information by changing the amplitude, frequency, or phase of the carrier signal. Figure 2 illustrates the signal generation process in a simplified zero intermediate frequency (IF) transmitter. Baseband modulation means the process of mapping from the bit stream to the symbol stream. Different modulation formats correspond to different symbol patterns; thus, modulation classification identifies the symbol patterns according to the received signals. Since the baseband signal can usually be expressed in plural form, the output of the modulation module is also called the baseband IQ signal. Up-conversion block refers to the process of moving the baseband IQ signal from the lower frequency to the carrier frequency. A digital-to-analog converter (DAC) transforms the digital signal into an analog signal, which is then mixed with two separate quadrature carriers and summed together. Before the synthesized RF signal is sent through the antenna, it has to be amplified to ensure that the signal can travel a sufficient distance.

The signal output from the baseband modulation module can be expressed as
(1)$$s^{i}\left(t\right)=\sum_{k=-\infty }^{+\infty }g\left(t-kT_{s}\right)S_{k}^{i},$$
where $s^{i}\left(t\right)$ is the time-domain signal expression of the *i*th modulation, g(t) is the time-domain response of the shaping filter, $T_{s}$ is the symbol period, and $S_{k}^{i}$ is the symbol sequence obtained using the *i*th modulation. In the up-conversion block, there is a certain degree of amplitude imbalance and carrier-phase non-orthogonality between the I and Q signals due to manufacturing defects of the DAC and the mixer. The signal output from up-conversion module can be expressed as
(2)$$\hat{s}\left(t\right)=\left(\mu s\left(t\right)+vs^{*}\left(t\right)\right)e^{j2\pi f_{c}t}$$
with
(3)$$\begin{array}{cc} \mu & =\frac{1}{2}\left(\alpha +1\right)cos\left(\frac{\pi \beta }{360}\right)+\frac{j}{2}\left(\alpha -1\right)sin\left(\frac{\pi \beta }{360}\right) \end{array}$$
(4)$$\begin{array}{cc} v & =\frac{1}{2}\left(\alpha -1\right)cos\left(\frac{\pi \beta }{360}\right)+\frac{j}{2}\left(\alpha +1\right)sin\left(\frac{\pi \beta }{360}\right) \end{array}$$
(5)$$\begin{array}{cc} \alpha & =20\log\frac{Gain_{I}}{Gain_{Q}}, \end{array}$$
where $\hat{s}\left(t\right)$ is the signal under the complex conjugate model and represents the RF signal generated by a quadrature modulator with IQ imbalance. s(t) is the IQ signal in Equation (Equation 1) without superscript, and $s^{*}\left(t\right)$ is the conjugate of s(t). $f_{c}$ is the carrier frequency. $Gain_{I}$ is the gain in the in-phase branch, $Gain_{Q}$ is the gain in the quadrature branch, and α denotes the amplitude imbalance between the two DACs. β is the extent to which the two orthogonal carriers deviate from orthogonality, expressed in radians. After being amplified by a non-ideal amplifier function PA[·], the signal sent by the antenna can be expressed as
(6)$$RF\left(t\right)=PA[\hat{s}\left(t\right)]$$

In wireless communication systems, since the carrier frequency is usually much higher than the modulated signal bandwidth, the nonlinearity of the amplifier can be approximated to be frequency independent. RF(t) can be written as
(7)RF(t)=PAμs(t)+vs*(t)ej2πfct(8)=PAμs(t)+vs*(t)ej2πfct

At the receiver side, the baseband signal after down-conversion can be expressed as
(9)y(t)=h(t)∗RF(t)e−j2πfLOt+θ0+n(t)(10)=PAμs(t)+vs*(t)ej2πfct·e−j2πfLOt+θ+n(t)(11)=PAμs(t)+vs*(t)ej2π(fc−fLO)t+θ+n(t)
where h(t) denotes the impulse response of the channel. In this work, for the purpose of simplifying the problem, we assume that the signal is only affected by Gaussian noise and the interference of channel variations on the detection effect is ignored, so we set the function h(t) to a constant. n(t) is the additive white Gaussian noise (AWGN) with zero mean and variance $\sigma^{2}$. $f_{LO}$ is the demodulation carrier frequency at the receiver, and θ is the demodulation carrier-phase offset. We define the signal-to-noise ratio (SNR) of the received signal as $\gamma =1/\sigma^{2}$. From Equation (11), it can be seen that the nonlinear mapping of the amplifier in the received signal acts equivalently on the baseband signal at the transmitter; therefore, the mapping process of the amplifier can be described using an equivalent baseband model.

To determine the form of the PA[·] function, we chose several memoryless amplifier models to simulate the nonlinear amplification behavior of different transmitter individuals. Meanwhile, we referred to the model parameter settings in the published literature [51,52,53,54] to approximate the real-device characteristics. To simplify the analysis, we assumed that the equivalent baseband signal of the frequency band signal input to the amplifier was
(12)$$\hat{s}\left(n\right)=r\left(n\right)\cdot exp\left[j\phi \left(n\right)\right],$$
where r(n) is the instantaneous amplitude, and ϕ(n) is the instantaneous phase; then, the output of the equivalent baseband amplifier model can be expressed as
(13)y(n)=A[r(n)]·exp[jϕ(n)+Φ(r(n))],
where A[·] and Φ[·] denote the AM/AM distortion and AM/PM distortion effects of the amplifier, respectively.

The Saleh model [51,52] can be used to describe the nonlinear characteristics of the traveling wave tube amplifier (TWTA), and it is widely used in the simulation of satellite communication systems. This model can be determined by the following four parameters $\alpha_{\alpha }$, $\beta_{\alpha }$, $\alpha_{\phi }$, and $\beta_{\phi }$, and its AM/AM and AM/PM response functions can be expressed, respectively, as
(14)$$\begin{array}{cc} A[r(n)] & =\frac{\alpha_{\alpha }r\left(n\right)}{1+\beta_{\alpha }r^{2}\left(n\right)} \end{array}$$
(15)$$\begin{array}{cc} \Phi [r(n)] & =\frac{\alpha_{\phi }r^{2}\left(n\right)}{1+\beta_{\phi }r^{2}\left(n\right)}. \end{array}$$

The Rapp model [53] is applied to solid-state power amplifiers (SSPA), and this model considers that the phase distortion of the signal is relatively small and therefore negligible. Its AM/AM and AM/PM response functions can be expressed, respectively, as
(16)$$\begin{array}{cc} A[r(n)] & =\frac{r(n)}{{\left[1+{\left(\frac{r(n)}{V_{sat}}\right)}^{2p}\right]}^{\frac{1}{2p}}} \end{array}$$
(17)$$\begin{array}{cc} \Phi [r(n)] & \approx 0, \end{array}$$
where $V_{sat}$ is the saturation output voltage of the amplifier, *p* is the smoothness factor, and the larger the *p* value, the more linearized the amplifier will be.

The CMOS model [54] is based on the Rapp model, which requires that its AM/AM characteristics obey the Rapp model criteria, while its AM/PM characteristics are also nonlinear rather than constant, which can be expressed as
(18)$$\begin{array}{cc} A[r(n)] & =\frac{r(n)}{{\left[1+{\left(\frac{r(n)}{V_{sat}}\right)}^{2p}\right]}^{\frac{1}{2p}}} \end{array}$$
(19)$$\begin{array}{cc} \Phi [r(n)] & =\frac{d\cdot r^{f}\left(n\right)}{1+{\left[\frac{r(n)}{e}\right]}^{g}}, \end{array}$$
where d,e,f, and *g* are the model parameters that control the degree of the phase nonlinearity.

## 4. AMSCN: AMC Mask-Based SEI Classification Network

This section is divided into two subsections. In the first, we discuss the overall training process of the AMSCN, as well as the objective function of the model. In the second section, we give the implementation details of the DenseNet part, the Transformer part, and the mask-based dual-head classifier (MDHC).

### 4.1. Offline Training Process

A supervised learning algorithm in deep learning aims to learn a mapping from the input to the output given a training set of inputs *x* and outputs *y*. Figure 3 provides the offline training process of our proposed AMSCN model.

As shown in Figure 1, in the data collection phase, we needed to choose each emitter in turn to send signals under each modulation type at the transmitter side. We denoted the received signal by $x^{IQ}$ and assumed that there were *U* modulation formats and *V* different transmitters; then, there would be U×V kinds of received-signal sets. Each set of the same kind was further split into multiple signal segments of length *L*, and the entire dataset, which contained *N* samples, can be represented as
(20)$$\left(\Omega_{x^{IQ}},y^{M},y^{E}\right)=\left\{\left(x_{1}^{IQ},y_{1}^{M},y_{1}^{E}\right),\cdot \cdot \cdot ,\left(x_{i}^{IQ},y_{i}^{M},y_{i}^{E}\right),\cdot \cdot \cdot ,\left(x_{N}^{IQ},y_{N}^{M},y_{N}^{E}\right)\right\},$$
where $\Omega_{x^{IQ}}$ is the sample set, y is the label set, $\left(x_{i}^{IQ},y_{i}^{M},y_{i}^{E}\right)$ is the *i*th (*i = 1, 2, ..., N*)-labeled sample in the entire dataset, and $y_{i}^{M}$ and $y_{i}^{E}$ are vectors encoded with one-hot to indicate the modulation and the transmitter to which this sample belongs.

We used the maximum likelihood estimation to find the best parameters Θ in the AMSCN. Since our framework contained two classification tasks, we let the model output the predicted probabilities under each classification task. For the AMC task, the probability $h_{\theta }^{M}\left(x^{IQ}\right)$ vector can be expressed as
(21)$$h_{\theta }^{M}\left(x^{IQ}\right)=\left[h_{\theta |M=i}\left(x^{IQ}\right)\right],\;\;i=1,2,...,U,$$
with
(22)$$\sum_{i=1}^{U}h_{\theta |M=i}\left(x^{IQ}\right)=1,$$
where *M* denotes the AMC task, and $h_{\theta |M=i}\left(x^{IQ}\right)$ is the probability that the current signal belongs to the *i*th modulation type. In this vector, the subscript of the largest probability is the subscript of the modulation method to which the current sample belongs, which is given by
(23)$$u=\underset{i}{\arg\max}h_{\theta |M=i}\left(x^{IQ}\right),$$
where *u* indicates the modulation format of the current signal. For the SEI task, the probability vector was generated according to the type of modulation predicted by the current model. In fact, the classifier of the SEI task initially output a set of *U* vectors.
(24)$$H_{\theta }^{E}\left(x^{IQ}\right)=\left\{h_{\theta }^{E|M=u}\right\},\;\;u=1,2,...,U,$$
where H is the ensemble of *U* sets of vectors, and $h_{\theta }^{E|M=u}$ is the representation of the probability that the signal belongs to each transmitter under a modulation assumed to be *u*. We selected one of these *U*-group vectors based on the results of the AMC classifier in the previous step, and used this vector as a distribution probability to characterize the source of the device to which the current signal belongs.

The goal of the AMSCN model is to optimize the two tasks simultaneously. In this process, the classification process of the AMC was relatively independent, while the classification process of the SEI needed the prediction results of the AMC to improve this task. The overall loss of the AMSCN was derived by summing the cross-entropy losses of the two tasks, namely
(25)$$L\left(\theta \right)=L^{M}\left(\theta \right)+L^{E}\left(\theta \right),$$
with
(26)$$\begin{array}{cc} L^{M}\left(\theta \right) & =-\frac{1}{N}\sum_{i=1}^{N}\sum_{u=1}^{U}y_{i}^{M}\log\left(h_{\theta |M=u}\left(x^{IQ}\right)\right) \end{array}$$
(27)$$\begin{array}{cc} L^{E}\left(\theta \right) & =-\frac{1}{N}\sum_{i=1}^{N}\sum_{v=1}^{V}y_{i}^{E}\log\left(h_{\theta |E=v}\left(x^{IQ}\right)\right), \end{array}$$
where $L^{M}\left(\theta \right)$ and $L^{E}\left(\theta \right)$ denoted the cross-entropy loss of the AMC and SEI, respectively. In the case of using the MDHC, we could achieve a good training effect without balancing the weight between $L^{M}\left(\theta \right)$ and $L^{E}\left(\theta \right)$.

### 4.2. Details of the AMSCN

As is shown in Figure 4, the AMSCN consisted of two modules, i.e., a shared-backbone module and an MDHC. The shared-backbone module was responsible for providing the distinguishable features for the following AMC and SEI tasks. To achieve better multitask classification, we subsequently used two different heads in the MDHC to map the common features into two task-specific features. Finally, the softmax function was used to convert these features into their respective class probability distributions, and the subscript of the maximum probability was used as the output of this classification task.

The shared-backbone module was responsible for extracting the common features needed for both tasks, and was subdivided into a DenseNet part and a Transformer part. The DenseNet part consisted of a cascade of several convolutional units of the same configuration, each of which was computed in the order of BatchNorm, ReLU activation, and one-dimensional convolutional operations. The more convolution units, the richer the nonlinear transformation of the signal; however, this increased the computational complexity. Unlike the application of DenseNet in image recognition, the data length of the signal was shorter than that of the image. In the SEI task, the max-pool operation corrupts minor characteristics in the signal; therefore, we did not use the transition layer as in [55]. In the DenseNet part of the AMSCN, the length of the input signal and the length of the output features were the same, which had the benefit of allowing shallow features to be directly cascaded with deep features by channel, thus increasing the scale diversity of the CNN output. The detailed parameter settings are shown in Table 1; the feature dimensions are represented using B×C×L, where *B* denotes the batch size, *C* denotes the channel size, and *L* denotes the feature length.

Since the modulation type and transmitter hardware impairments affect the waveform of the signal at different scales, a large convolution kernel 1×11 was used in our base convolution unit instead of the traditional small convolution kernel to increase the network’s ability to capture features at different scales. In addition, since we did not use pooling operations to reduce the feature dimension, the computational effort grows exponentially with more convolutional layers, so we fixed the number of channels output from each convolutional layer to 16 and used cross-channel cascading to increase the channel richness of the final output features. The number of feature channels output from the DenseNet part is equal to the sum of the output channels of each convolution unit and the original 2 channels of the signal, i.e., 162.

After the local features were extracted by the convolution module, we used the Transformer structure to improve the relevance of the information in the features with different time spans. Since the features were not shortened in the DenseNet part, the Transformer structure was well suited for processing such long span features. In the AMSCN, the Transformer module mainly consisted of two transformer blocks, and the complete structure is shown in Figure 5.

In the figure above, we used X to denote the multichannel features output by the convolution module. Since the features on each channel had the same length, we grouped the features with the same sequence position on each channel and named them as a node. X is a sequence of nodes of length *L*, and the vector length of each node is *C*. Since the Transformer does not consider the relationship order between nodes during the computation, we needed to encode the position relationship of the nodes into the input sequence X before the first transformer layer. In the AMSCN, we set a learnable parameter sequence with the same dimension as X, i.e., position embedding, and added it directly to X to obtain the encoded feature sequence. Before the position embedding, we needed to add an additional node to X. This node was separated for the classification vector after the two-layer transformer computation was completed.

A transformer block consists of two main components: self-attention and feed-forward. The outputs of both components were connected using residuals to enhance the backpropagation of the gradient. The self-attention part was the core of the transformer block, whose structure is shown in Figure 6.

A multi-head attention module can be seen as a composition of multiple single-head scaled dot-product attention modules. The calculation of scaled dot-product attention can be expressed as
(28)$$\mathrm{Attention}\left(Q,K,V\right)=softmax\left(\frac{{\mathrm{QK}}^{T}}{\sqrt{d_{k}}}\right)V,$$
where Q is a matrix representation of the query vector, K is a matrix representation of the key vector, V is a matrix representation of the value vector, and $d_{k}$ is the dimension of the input feature, which can be used to scale the ${\mathrm{QK}}^{T}$; softmax(·) is the activation function to ensure that the output vector value is between 0 and 1. The calculation of the multi-head attention can be expressed as
(29)$$\begin{array}{c} \mathrm{MHA}\left(Q,K,V\right)=W^{O}Concat\left(h_{1},h_{2},\cdot \cdot \cdot ,h_{m}\right) \end{array}$$
(30)$$\begin{array}{c} h_{i}=\mathrm{Attention}\left(W_{i}^{Q}Q,W_{i}^{K}K,W_{i}^{V}V\right), \end{array}$$
where $W^{O}$ is the linear transformation matrix of the multi-head attention result, $h_{i}$ is the vector of a dot-product attention calculation, and *m* is the number of multi-heads. $W_{i}^{Q}$ is the linear transformation matrix of the input query matrix Q, and $W_{i}^{K}$ and $W_{i}^{V}$ have a similar meaning. In the AMSCN, there were two heads in each multi-head attention module. The feed-forward was a two-layer fully connected layer; the first layer contained the ReLU activation function and had 1024 neuron nodes, and the second layer did not use the activation function, which can be expressed by the following equation
(31)$$\begin{array}{c} \mathrm{FF}=W_{2}\mathrm{Max}\left(0,W_{1}X+b_{1}\right)+b_{2}, \end{array}$$
where Max(·) denotes the ReLU activation, X is the input to the feed-forward module, $W_{1}$ and $W_{2}$ are the linear transformation matrices of the first and second layers, respectively, and b1 and b2 are the biases of each layer.

The second transformer block, shown in Figure 5, had exactly the same parameters as the first transformer block; the stacked transformer blocks can achieve a more complex synthesis of temporal information. We took out the first node of the sequence output by the second transformer block, which was the classification token, as the semantic feature of the shared-backbone network output. A more specific hyperparameter setting is shown in Table 2.

To implement a network model that performs two classification tasks, we connected the MDHC to the shared-backbone network. The structure of the two classifier heads in the MDHC was basically the same, except for the setting of FC2. For each classifier head, the number of the FC1 layer’s hidden neurons was 256. The length of the feature vector output from the AMC header was equal to the number of modulation types *U*, while the length of the feature vector output from the SEI header was equal to U×V. We reshaped this feature vector into a *U*-row and *V*-column feature matrix P and generated the mask matrix Q based on the AMC feature vector. The dimension of the mask matrix Q was U×V. We note that the subscript of the maximum value in the AMC eigenvector was *k*. Subsequently, the *k*th row of the mask matrix was set to 1, and the remaining rows were all set to 0. Finally, we summed the matrix P·Q by columns to obtain the final SEI feature vector. Figure 7 gives a schematic representation of the calculation process when U=3 and V=3.

Generally speaking, each classifier of multi-task learning is independent of each other; however, in this paper, we need the decision results of the AMC task to have an impact on the decision process of SEI. In MDHC, we can consider that there are multiple SEI classifiers existing simultaneously, each of which predicts a classification result. We picked one of the results based on the prediction of AMC. If the prediction of the AMC task fails, then to a large extent, the SEI results will also be unreliable. At this point, the loss value of the network rises, thus forcing the model to first learn the modulation features and then use them to assist in adjudicating the SEI task.

## 5. Results and Analysis

### 5.1. Dataset Generation and Training

Since there was no publicly available dataset suitable for the problem at hand, we generated a simulation dataset and verified the effectiveness of the AMSCN on this dataset. We selected five types of digital modulations, which were BPSK, QPSK, 8PSK, 16QAM, and 32QAM. At the same time, we also set the parameters of the five transmitters with different amplitude imbalance, phase imbalance, and nonlinear power amplifier behaviors. The parameter settings are shown in Table 3.

The symbols in this table are consistent with Equations (Equation 3)–(Equation 18). The term *Limits* means there were some restrictions on the signal before it entered the PA model. E$\left\{|r{\left(n\right)}^{2}|\right\}$ indicates the average power of the input signal to be satisfied, and max(|r(n)|) means that the amplitude of each IQ point of the input signal was limited to 1. We found the values of these amplifier parameters from the literature mentioned in Section 3.2, which were derived from measurements and fits to real amplifier devices.

We generated the baseband signal according to the process shown in Figure 2, with a roll-off factor of 0.25 for the raised cosine filter at the transmitter and a signal-to-noise ratio (SNR) from −20 to 20 dB in steps of 4 dB. At the receiver side, the number of points sampled per symbol was 8, and 625 signal segments were acquired for each combination of modulation, device, and signal-to-noise ratio, with each segment having a length of 256 samples. Thus, the total number of signal segments received in the dataset was 5×5×11×625 (5 transmitters, 5 modulations, 11 SNRs, and 625 segments for each condition). We randomly shuffled the above dataset and sliced it into training and testing sets in the ratio of 6:4. In the offline training phase, the Adam optimizer [56] was used to search for the model parameters, and the training process was terminated when the average classification accuracy on the testing set no longer increased.

### 5.2. The Effect of the Mask between the Two Heads

The initial idea of this paper was to fuse the AMC task and the SEI task within one single model to save network parameters and improve the computational efficiency. Therefore, we let the two tasks share the same backbone network, and let the two classifier heads be responsible for their respective prediction tasks. In general, different modulation methods have a greater impact on the signal waveform compared with different device hardware impairments. The design methodology of the MDHC was based on this assumption. When the two tasks occur at the same time, the change in modulation type may interfere with the extraction of the transmitter features from the signal. Therefore, we let the SEI head predict five sets of results simultaneously, and each set could be viewed as the conditional probability of the five transmitters with a known modulation method. We selected the appropriate one from the five sets of results output from the SEI head based on the predicted results from the AMC head. This process was similar to the AMC head, generating a mask that covered the output results of the SEI head. Figure 8 compares the effect of using and not using the mask operation.

We used the average accuracy to measure the difference in performance between the different methods. For a given task, the average accuracy is the average probability of being able to classify all the samples of the test set correctly, and the average accuracy can be seen as the average of the accuracies under each SNR in the graph. In the AMC task, the average accuracy of using the mask method was 0.65, and it was 0.6327 without the mask; in the SEI task, the average accuracy of using the mask was 0.5472, and it was 0.5413 without the mask.

### 5.3. The Effect of the Fusion of the Two Tasks

In order to demonstrate that the fusion of two tasks in the AMSCN could ensure the optimal result of both tasks, the following comparative experiments were conducted. We trained a single-head model for the AMC task and a single-head model for the SEI task. The structure of the single-head model was similar to that of the AMSCN, except that each single-head model used one classifier head to predict the current task without considering the possible information gain from the other task. In Figure 9, we use the term Fusion to refer to our AMSCN model, and Single to refer to the model with one task head. In the AMC task, the average accuracy of the fusion method was 0.65, and it was 0.651 for the single method; in the SEI task, the average accuracy of fusion method was 0.5472, and it was 0.5142 for the single method.

In Figure 9a, the accuracy curve of the single-task model was slightly higher at high SNRs than that of the dual-task model, which we believe is mainly caused by setting the average accuracy as the optimization target. During model training, the optimization weights were the same for each SNR, but the accuracy curve may fluctuate in the range of high and low SNRs when the average accuracy is similar. Specifically, in the early stage of model optimization, the accuracy under low SNR and high SNR would increase simultaneously, while at the end of model convergence, the accuracy under high SNR would decrease if the model gradually advances toward improving the accuracy under low SNR. This observation may be caused by the conflicting judgment methods between the signal feature vectors of various modulation types at low SNRs and those at high SNRs, which makes it difficult for the model to improve the accuracy at each SNR simultaneously. In addition, the structure of the AMSCN showed that the model was not affected by the predicted results of SEI under the AMC task, and thus the dual task maintained an essentially similar performance to the single task. The reason for our design is that we argue the judgment difficulty of the AMC task is lower than that of the SEI, i.e., the hardware impairments of the transmitter have less impact on the signal characteristics than the modulation type, and thus the detection results of the AMC task do not suffer from the SEI task. Therefore, the advantage of our model is not to improve the performance of AMC task, instead, we exploit the strength of multi-task learning to improve the performance of SEI tasks while maintaining the performance of AMC.

### 5.4. The Performance Comparison among Different Methods

We selected eight popular signal classification models to compare with our AMSCN model. Some of these models have achieved state-of-the-art results with publicly available AMC or SEI datasets [57,58]. The ResNet12 is a simplified version of ResNet18 [59], and the DenseNet-backbone is the same as the DenseNet part of the AMSCN. All models were trained and tested in a single-task classification manner only. However, since the length of our sample data is 256, which is different from the length of the data on which these models were designed, the training results tend to be poor when the hyperparameters are set according to the original version; therefore, we fine-tuned the hyperparameters of these models. These adjustments included changing the original classifier hidden layer unit of 128 to 512, and the original LSTM structure’s hidden unit of 128 to 256 or 512. After the adjustments, the prediction accuracy results are generally better than the original parameters.

Figure 10 gives a comparison of the SNR performance curves between the models, and Table 4 lists the detailed average accuracy of each model. On the AMC task, the AMSCN remained at the same level as other models, and we had the best results in terms of average accuracy; however, in terms of each SNR, they were not optimal. On the SEI task, we were ahead of other models and achieved the best results at almost all SNRs. As can be seen from Table 4, a model that performs better on one task may perform worse or not converge on the other task, which indicates that there is variability, such as differences in feature scales, between the two tasks. For the AMSCN, there are two design considerations that enable our model to be compatible with both tasks: the design of a more compatible backbone network and the design of a mechanism (MDHC) for transferring relevant information between tasks.

By taking advantage of multi-tasking, we can reduce the computation of both models into one model, thus improving the detection efficiency of the tasks. Table 5 gives the number of parameters, the amount of floating point calculations, and the single-inference time for each model. These tests were performed on the same hardware and software environment, including Pytorch 1.12.1, Intet(R) Xeon CPU E5-2620, and one GeForce RTX 2080Ti. As can be seen from Table 5, although AMSCN is not the smallest in terms of number of parameters and computational effort, it benefits from the better parallel performance of the convolution and Transformer structures in the network structure, resulting in a shorter single-inference time compared with the model using the LSTM structure.

It should be emphasised that the computational complexity of the other models is measured under a single task, whereas the AMSCN is under a dual task; therefore, the computational complexity of all the other models should be doubled for a system containing two tasks. If we consider the single processing time as an important indicator of complexity, we can say that the dual-task-based AMSCN approach achieves the best results in terms of the combination of accuracy and efficiency metrics.

Our model has the advantage of good parallelism, thus maintaining a high computational speed in high-capacity backbone networks; however, the storage overhead is high and is not suitable for cost-sensitive computing platforms. Therefore, compressing the size of the model to further increase the computational speed while keeping the performance constant will be our next research goal.

## 6. Conclusions

In this paper, we presented the problem of possible efficiency loss in conventional RF signal-monitoring systems, where the same signal is passed through multiple independent detection systems for achieving multiple related detection targets. Specifically, in the scenario where both AMC and SEI tasks are required, we proposed a novel multitask classification method, named an AMSCN, to obtain gains in detection accuracy or computational efficiency. The AMSCN learned the common features of two tasks simultaneously through a high-capacity backbone network, and then learned the unique features of each task separately through the mask-based dual-head classifier. Simulation experimental results showed that AMC and SEI are task-dependent, and the dual-task model can take full advantage of this correlation and improve the detection accuracy of SEI. On the AMC task, the detection accuracy of the AMSCN remained at the same level as the state-of-the-art model, while on the SEI task, there was a 2.5% improvement compared with the state-of-the-art model. In addition, the computational structure of the AMSCN was the same under both tasks, which can simplify the training and deployment of monitoring systems under multiple tasks. In conclusion, we believe that multi-task learning has great potential in the field of signal monitoring, and the more comprehensive the information obtained by the model during training, the more accurate the recognition of the signal will be. In the future, we will further explore the applicability of the AMSCN method, improving its robustness under different channels, detecting more modulations, and fusing more detection targets in the same model to improve efficiency.

## Figures and Tables

**Figure 1 sensors-23-02476-f001:**
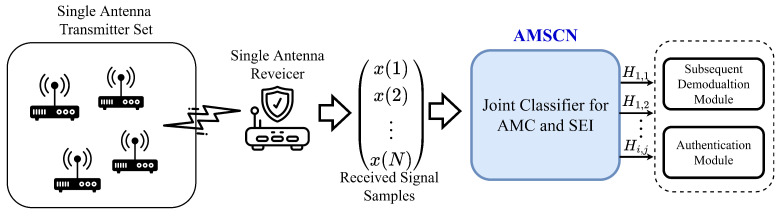
The modulation and transmitter joint-identification framework.

**Figure 2 sensors-23-02476-f002:**
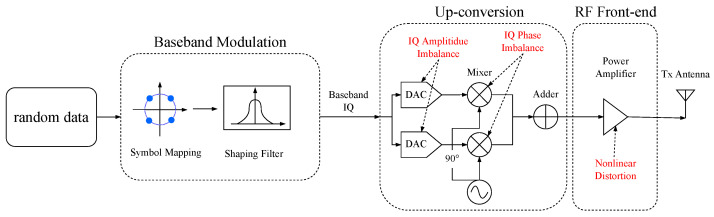
The process of digital modulation and up-conversion.

**Figure 3 sensors-23-02476-f003:**
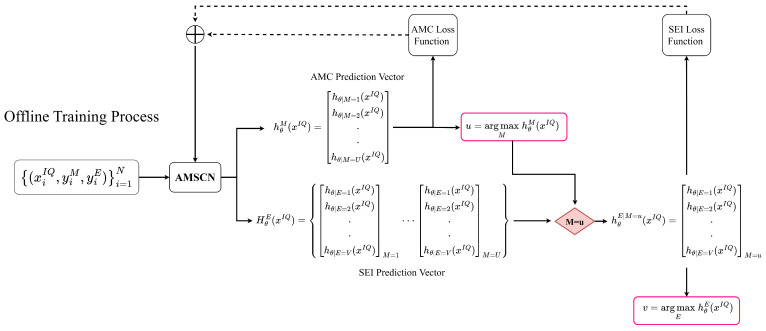
Offline training process of the proposed AMSCN.

**Figure 4 sensors-23-02476-f004:**
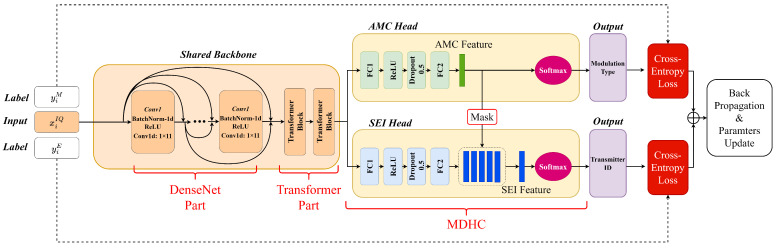
Schematic diagram of forward and backward propagation of AMSCN.

**Figure 5 sensors-23-02476-f005:**
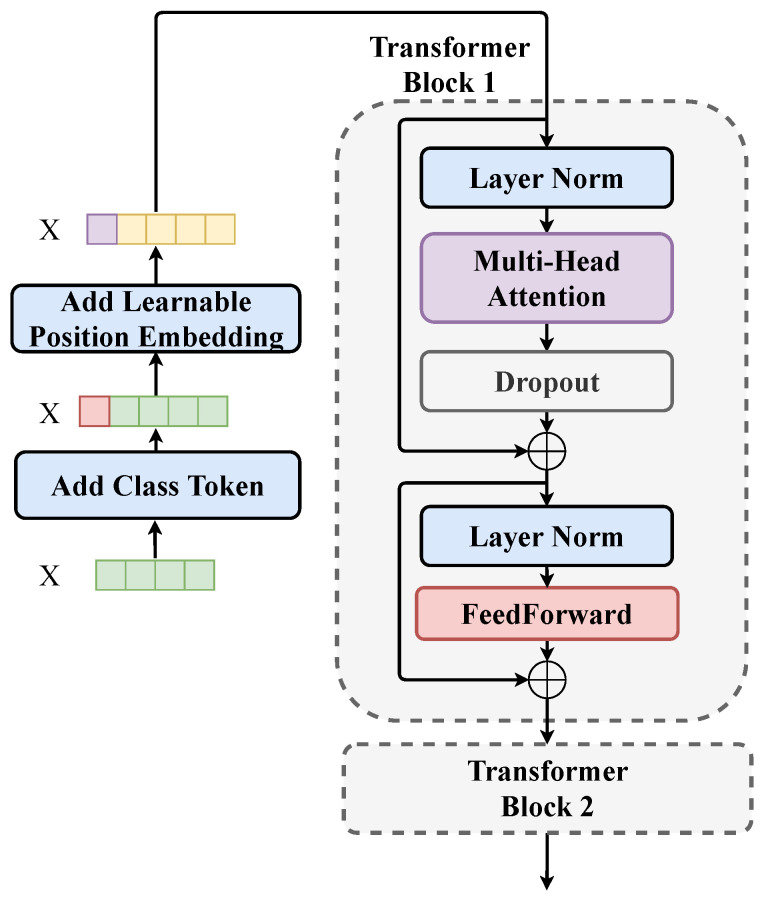
Schematic diagram of the Transformer part of the AMSCN. Before the features are fed into the first transformer block, preprocessing is required, i.e., adding the classification token and position embedding.

**Figure 6 sensors-23-02476-f006:**
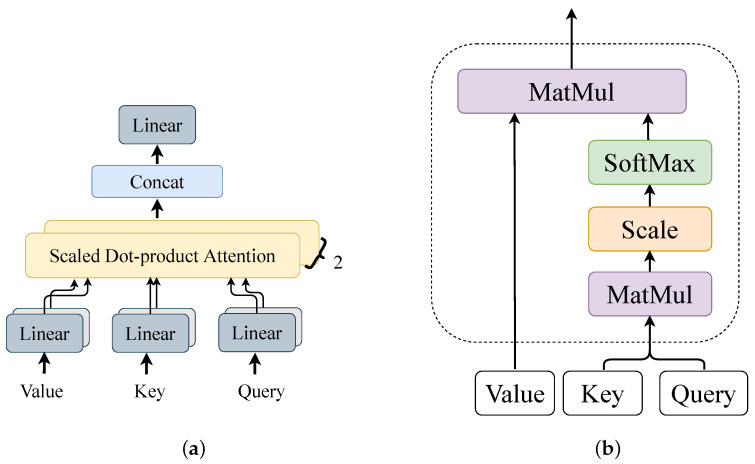
The structure of the self-attention part a transformer block. (**a**) Details of the multi-head attention structure. (**b**) Scaled dot-product attention structure.

**Figure 7 sensors-23-02476-f007:**
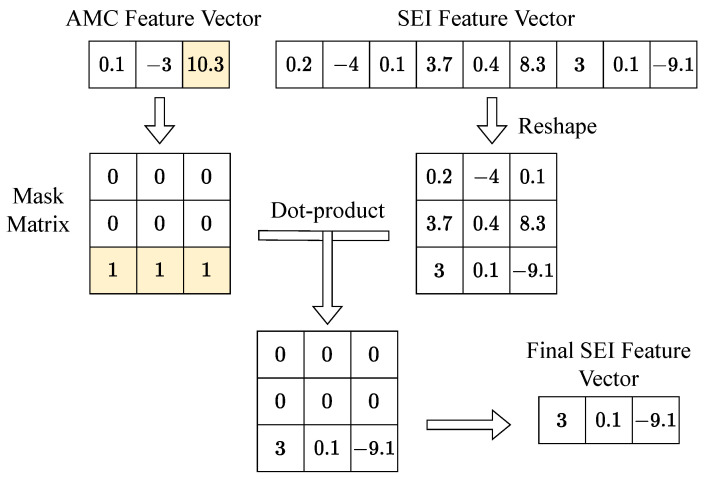
The calculation process of the SEI feature vectors. The mask matrix is generated from the AMC feature vector. The subscript of the largest element of the AMC feature vector determines which row of the mask matrix is 1. The colored part of this figure indicates the maximum value in the vector and the row of the matrix that takes 1.

**Figure 8 sensors-23-02476-f008:**
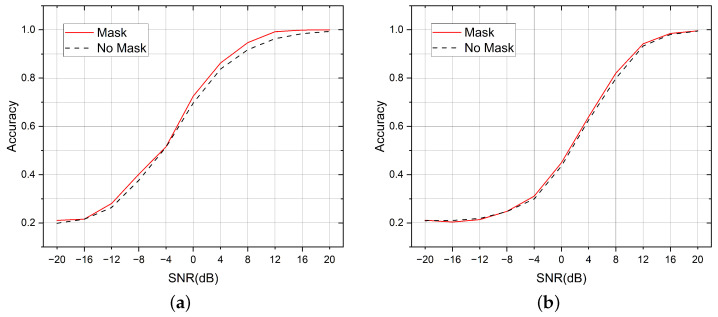
Adding a mask between the two task headers improves the performance on both tasks. (**a**) Comparison of the mask effects in the AMC tasks. (**b**) Comparison of the mask effects in the SEI tasks.

**Figure 9 sensors-23-02476-f009:**
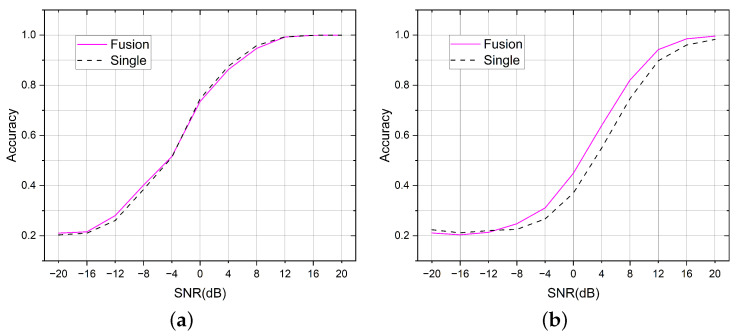
For the AMSCN models, multitask training is more effective than single-task training. (**a**) Comparison of the fusion method and the single-head method in the AMC tasks. (**b**) Comparison of the fusion method and the single-head method in the SEI tasks.

**Figure 10 sensors-23-02476-f010:**
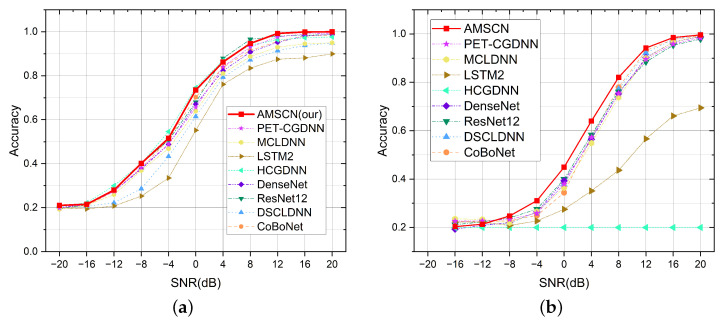
Performance comparison between the AMSCN and other popular models. (**a**) Model performance curves in the AMC tasks. (**b**) Model performance curves in the SEI tasks.

**Table 1 sensors-23-02476-t001:** The hyperparameters of the DenseNet part of the AMSCN.

Input: $x^{\mathrm{IQ}}$ (Dimension: B×2×L)				
**Layers**	**Kernel Size**	**Padding**	**Stride Step**	**Output Feature Dimension**
Conv1	16@(1×11)	5	1	B×(16+2)×L
Conv2	16@(1×11)	5	1	B×(16×2+2)×L
Conv3	16@(1×11)	5	1	B×(16×3+2)×L
...	...	...	...	...
Conv10	16@(1×11)	5	1	B×(16×10+2)×L
Output: Feature Matrix (Dimension: B×162×L)				

**Table 2 sensors-23-02476-t002:** The hyperparameters of the two Transformer blocks.

Input: $x^{\mathrm{IQ}}$ (Dimension: B×162×L)		
**Layers**	**Description**	**Output Feature Dimension**
Reshape	Change the order of data dimensions	B×L×162
Add Class Token	Increase sequence length	B×(L+1)×162
Add Position Embedding	No change in data dimension	B×(L+1)×162
Layer Norm1	Channel-by-channel normalization	B×(L+1)×162
Multi-Head	Fully connected structure, 2 MHA groups, output merging	B×(L+1)×162
Dropout	Drop rate 0.4	B×(L+1)×162
Residual connection	Add operation	B×(L+1)×162
Layer Norm2	Channel-by-channel normalization	B×(L+1)×162
FeedForward	Two layers fully connected, hidden cell 128, drop rate 0.4	B×(L+1)×162
Residual connection	Add operation	B×(L+1)×162
The second transformer block	Same configuration as the first transformer block	B×(L+1)×162
Extracting class token	The first vector in the sequence	B×1×162
Output: Feature Matrix (Dimension: B×1×162)		

**Table 3 sensors-23-02476-t003:** Parameter settings for the five different transmitters.

Device	Amplitude Imbalance α(dB)	Phase Imbalance β(°)	PA Model	Parameters	Limits
Device1	−0.5	−10	Saleh [51,52]	$\alpha_{\alpha }$ = 1.2, $\beta_{\alpha }$ = 0.36, $\alpha_{\phi }$ = 0.374, $\beta_{\phi }$ = 0.36	E$\left\{|r{\left(n\right)}^{2}|\right\}$ = 0.5, max(|r(n)|)≤ 1
Device2	−0.3	−6	Rapp [53]	$V_{sat}$ = 2, *p* = 1	E$\left\{|r{\left(n\right)}^{2}|\right\}$ = 0.6, max(|r(n)|)≤ 1
Device3	−0.1	−2	Saleh [51,52]	$\alpha_{\alpha }$ = 1.9638, $\beta_{\alpha }$ = 0.9945, $\alpha_{\phi }$ = 2.5293, $\beta_{\phi }$ = 2.8168	E$\left\{|r{\left(n\right)}^{2}|\right\}$ = 0.5, max(|r(n)|)≤ 1
Device4	0.1	2	CMOS [54]	$V_{sat}$ = 0.81, *p* = 0.58, *d* = 44.68, *e* = 0.114, *f* = 2.4, *g* = 2.3	E$\left\{|r{\left(n\right)}^{2}|\right\}$ = 1.162, max(|r(n)|)≤ 1
Device5	0.3	6	Saleh [51,52]	$\alpha_{\alpha }$ = 2.1587, $\beta_{\alpha }$ = 1.1517, $\alpha_{\phi }$ = 4.0033, $\beta_{\phi }$ = 9.1040	E$\left\{|r{\left(n\right)}^{2}|\right\}$ = 0.5, max(|r(n)|)≤ 1

**Table 4 sensors-23-02476-t004:** The average accuracy comparison between the AMSCN and other popular models.

Model	AMC	SEI
AMSCN (ours)	0.650	0.547
HCGDNN [60]	0.649	0.2
ResNet12	0.642	0.519
PET-CGDNN [61]	0.631	0.520
CoBoNet [58]	0.631	0.522
DenseNet (backbone)	0.625	0.513
MCLDNN [62]	0.606	0.518
DSCLDNN [63]	0.584	0.521
LSTM2 [64]	0.545	0.372

**Table 5 sensors-23-02476-t005:** Comparison of computational efficiency between models. The metric “Latency” stands for the inference time for a single signal sequence. If the two tasks are considered together, the weight and the calculated amount of all other models should be doubled.

Model	Weight	Flops	Latency
AMSCN (ours)	1.57 M	259.34 M	0.007 s
HCGDNN	0.457 M	92.53 M	0.012 s
ResNet12	13.85 M	1549.2 M	0.006 s
PET-CGDNN	0.841 M	206.3 M	0.009 s
CoBoNet	0.67 M	135.3 M	0.035 s
DenseNet(backbone)	0.174 M	33.9 M	0.004 s
MCLDNN	3.53 M	872.6 M	0.007 s
DSCLDNN	1.13 M	293.2 M	0.014 s
LSTM2	3.16 M	811.08 M	0.006 s

## Data Availability

The dataset in the paper can be accessed from the following link https://github.com/WTI-Cyber-Team/Public_Wireless_Signal_Datasets (accessed on 5 January 2023).

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
