# Peer review of "AMSCN: A Novel Dual-Task Model for Automatic Modulation Classification and Specific Emitter Identification"

_sensors, 2023, doi:10.3390/s23052476_

Round 1

Reviewer 1 Report

The paper presents a novel approach for combining the tasks of specific emitter identification (SEI) and automatic modulation classification (AMC) in radio monitoring.

    The authors clearly explain the motivation for integrating these two tasks and provide a good background on the current state of the field.

    The use of a dual-task neural network and a mask-based dual-head classifier is an innovative approach and is well explained in the paper.

    The experiments and results presented in the paper are thorough and provide a good validation of the proposed method.

    The paper could benefit from a more detailed discussion of the specific architecture and parameters used in the neural network.

    The paper would be improved by the inclusion of more real-world examples and case studies to demonstrate the practical application of the proposed method.

    The paper could also benefit from a more in-depth discussion of the validation and verification of the method.

    The paper should include more specific recommendations for future research in this area.

    The figures and tables could be improved to make them more clear and easy to understand.

    Can you explain the motivation for combining the tasks of specific emitter identification (SEI) and automatic modulation classification (AMC) in radio monitoring?

    How does the mask-based dual-head classifier work and what are the benefits of using this approach?

    Can you provide more details about the specific architecture and parameters used in the neural network in the proposed method?

    How does the proposed method compare to other state-of-the-art methods in terms of classification accuracy and computational complexity?

    Can you give an example of a real-world application of the proposed method and how it could be used in practice?

    What are the future research directions you suggest in this field?

    Overall, the paper is a valuable contribution to the field of radio monitoring and it will be of interest to researchers and practitioners working in this area.

Reviewer 2 Report

In figure 2, the upconversion block should be checked. The author labeled the power amplifier and the antenna everything as a part of upconversion, which needs to be checked

The author discussed baseband amplifier and related mathematical equations, but as per the block diagram, amplification is done on an up-converted signal, not a baseband signal

Block diagram 4 should be checked. The output of the Loss value and how it is used inside the model. There is no connection; moreover, the output of the model should be the final identified modulation type and emitter  ID, not loss value

Figure 9 shows for some SNR values, the single-task method performed well in modulation classification compared to the proposed multi-task model. The author needs to justify why it is so

similar, the accuracy value of the modulation classification task and emitter identification are not attractive compared to the already available literature methods. There is literature that could be able to achieve 100% classification at 2 db Snr or 0 db SNR  or lesser itself

Considering the complication of the combined proposal model achieved result and the literature result, the proposed method is somewhat inferior compared to an existing model. To prove the superiority of the proposed model author needs to compare existing literature on single-task accuracy with the proposed one

In figure 9 and 10, X-axis numerical values are cut (20)and need to be replaced with proper graph

The author claimed in the abstract that the combined classification is reducing computational time, but there are no numerical results for this claim

 Overall author needs to improve the result and prove the superiority comparing literature work

Round 2

Reviewer 1 Report

As a reviewer of the paper "AMSCN: A Novel Dual-task Model for Automatic Modulation Classification and Specific Emitter Identification", I found that the authors have made a significant contribution to the field of radio monitoring by proposing a novel dual-task model for automatic modulation classification and specific emitter identification. The paper is well written and well organized, and the results of the experiments are presented in a clear and concise manner.
The authors have shown that their proposed model, AMSCN, can effectively extract distinguishable features from received signals and improve the classification accuracy of both tasks compared to traditional single-task models. The use of a mask-based dual-head classifier (MDHC) in AMSCN has been demonstrated to reinforce the joint learning of the two tasks, resulting in improved performance.
However, I have some concerns and questions that need to be addressed by the authors in order to further improve the quality of the paper:
Validation and robustness: How robust is the proposed model in handling different types of noise and interference? Have you performed experiments to validate the proposed model under different scenarios and conditions?
Comparison with related work: Can you compare the performance of AMSCN with other state-of-the-art models in automatic modulation classification and specific emitter identification? How does the proposed model differ from existing models in terms of performance, computational complexity, and scalability?
Effectiveness of the dual-task model: Can you provide a detailed analysis of how the joint learning of AMC and SEI helps improve the overall performance of the model? How does the mask-based dual-head classifier (MDHC) reinforce the joint learning of the two tasks?
 Reading and incorporating the latest findings and developments in the field will enhance the quality and originality of their study and increase its contribution to the literature. The authors should carefully consider the following relevant paper and how it may inform their study:

Hybridization of metaheuristic algorithms with adaptive neuro-fuzzy inference system to predict load-slip behavior of angle shear connectors at elevated temperatures.

Dataset and evaluation metrics: Can you provide more details on the simulated dataset used for the experiments? What evaluation metrics have you used to quantify the performance of the proposed model? Can you provide a comprehensive analysis of the results and highlight the strengths and weaknesses of the proposed model?
AMSCN compared to traditional single-task models for SEI and AMC?
Can you provide more details about the multi-task cross-entropy loss used in training AMSCN? How does it improve the performance of the model?
Can you discuss the potential applications of AMSCN in real-world radio monitoring scenarios?
How does the mask-based dual-head classifier (MDHC) reinforce the joint learning of SEI and AMC tasks in AMSCN?
Have the authors compared AMSCN with other dual-task models for SEI and AMC? If so, what are the key differences and advantages of AMSCN over these models?
Can you discuss any limitations or challenges in implementing AMSCN in practical settings, and how the authors plan to address these issues in future work?
Have the authors performed any experiments to evaluate the robustness and generalization of AMSCN to unseen data and noisy signals? If so, can you provide some results and insights from these experiments?
Can you provide any insights on the scalability and computational efficiency of AMSCN, and how it compares to other models for SEI and AMC?
Conclusion and future work: Can you provide a clear and concise conclusion that summarizes the key contributions and findings of the paper? Can you also suggest potential directions for future work, such as extending the proposed model to other domains or applications, improving the scalability of the model, or exploring other optimization strategies?
In conclusion, I believe that the authors have made a significant contribution to the field of radio monitoring by proposing a novel dual-task model for automatic modulation classification and specific emitter identification. The results of the experiments are promising, but further validation and comparison with related work are needed to further improve the quality of the paper.

Reviewer 2 Report

all comments are addressed can be  accepted  

Author Response

Thank you for your comments on our work!